# Inter-Work and Ethical Vigilance: Two Scenarios for the (Post-)Pandemic Future of Systems Thinking

**José-Rodrigo Córdoba-Pachón**

School of Business and Management, Royal Holloway, University of London, Egham Hill, Egham, Surrey TW200EX, UK; j.r.cordoba-pachon@rhul.ac.uk

**Abstract:** For several decades, systems thinking has been a defined body of knowledge that has contributed to many areas of science. Its value has, critically, resided in (meta- or post-) paradigmatic and participative use of one or several systems approaches to help stakeholders' structure and tackle complex problems. With renewed and (post-)pandemic interest in interdisciplinary work, this paper argues that to continue securing a future, system thinking requires a wider understanding of the dynamics and intertwining of knowledge unfolding and ethics in society. Two different but overlapping scenarios for systems thinking are proposed: (a) One based on inter-(disciplinary, para/professional, group) work and (b) another based on ethical vigilance. The first one is not so different from what has been envisaged for systems thinking in the last few years. Nevertheless, and following the ideas of the sociologist Andrew Abbott, this scenario proposes the explicit inclusion of the goal of knowledge rediscovery to promote a sense of solidarity, mutual understanding and compassion. For the second scenario, Michel Foucault's notion of governmentality is used to problematize pandemic events and practices, and to offer possibilities for individual critical thinking and action, also leading us to consider the importance of (self-other) compassion. Features, implications, questions and examples of use are provided for each scenario.

**Keywords:** systems thinking; approaches; Andrew Abbott; knowledge rediscovery; Michel Foucault; governmentality; coronavirus; pandemic; ethics

## 1. Introduction

In the last few decades, systems thinking has become a body of knowledge whose value has, critically, resided in, meta- or post-paradigmatic and participative use of one or several of these elements to help stakeholders structure and tackle complex situations. Under different names (i.e., soft operational research, complexity, applied systems thinking, cybernetics, etc.), this body of knowledge serves academic and practical audiences in several realms, including, but not limited to, operational research (OR), information systems, community relationships and health or environmental management [1].

The idea of a situation as a complex system has gained new prominence in the current climate of uncertainty stemming from the world pandemic [2–4]. As human science and its different knowledge actors strive to find solutions to the new and emerging challenges posed by this situation, issues like effective contact tracing, vaccine generation or physical distancing are now at the center of interdisciplinary and inter-professional work, both the natural and social sciences. To these and other challenges and a surrounding sense of urgency to find answers in the new "normal", a key question emerges: What can systems thinking do to continue offering value to academic and practical audiences?

This paper argues that addressing the above question requires systems thinkers to better understand the wider dynamics and intertwining of knowledge and ethics in societies. Under a proposed role of knowledge brokering, two different but overlapping scenarios for thinking and action

are thus proposed: (a) One based on inter-(disciplinary, professional, group) work, and (b) another based on promoting continuous ethical vigilance. The first one might not be so different from what has been envisaged for systems thinking in the last few years. Nevertheless, and following the ideas of the sociologist Andrew Abbott [5–8], this scenario proposes the explicit inclusion of possibilities for knowledge rediscovery to facilitate dialogue and interaction between the different stakeholder groups involved or marginalized in (post-)pandemic situations. This richer view of knowledge and social relations unfolding could bring a renewed sense of solidarity, mutual understanding and compassion between stakeholders.

The second scenario is inspired by Michel Foucault's notion of governmentality [9,10], given that inter-work in societies could also be constraining the power of human creativity to deal with (post-)pandemic situations. It becomes essential to promote continuous reflection on how creativity has become and is currently being organized in our societies [11–13], and what can be done within power relations to maintain a continuous sense of ethical vigilance, thinking and action in our relations with ourselves and others [14].

In this paper, the scenarios' features, implications, guiding questions and practical examples are detailed with the aim of helping systems thinkers and others, both in academia and practice, to better engage with relevant audiences and offer valuable insights and support to understand and deal with emerging situations of our world pandemic. With these, it is also proposed that the two scenarios become complementary: Inter-work would need to be continuously "invigilated", so that constraints and possibilities for the ethical flourishing of human creativity are drawn and put into practice.

The paper is structured as follows. The question of knowledge relevance is introduced by revisiting how systems thinking could contribute to generating valuable knowledge in different realms of life, albeit not free of tensions and debates. This leads us to identify a role of knowledge brokering for systems thinkers which they can thus better exert with consideration of the proposed scenarios. Scenarios are defined with features, implications and examples, and concluding remarks.

## 2. The Future(s) of Operational Research (OR) and Systems Thinking

It was systems thinkers like Ackoff and Churchman—who in the 1970s—, raised concerns about the usefulness of operational research (OR), and how systems thinking ideas could help OR researchers define and work on the *right* problems to solve [15,16]. At the time, these authors encouraged researchers, policy makers and managers to ensure that their activity had an explicit ethical commitment to secure improvements for present and future generations. This required enhancing the inter-disciplinary work in the sciences, which had already started during the Second World War in countries like the US and the UK.

For Churchman [15,17], systems thinking was a philosophically informed way of enabling people to reflect on the implications of their decisions by considering the assumptions they made about possible impacts in the future. This put systems thinking at the heart of processes of formulating and solving societal problems on a large scale. To date, Churchman's work on the design of inquiring systems and further developments on systems boundary critique can provide forms of dialectical inquiry to help stakeholders assess if they are solving the right problem or sets of problems, as well as reflect on the ethical values that guide their thinking and action.

Since the 1970s and following Churchman, several developments have taken place, in particular those of approaches to facilitate problem identification and problem solving. Currently, many of these could be seen as constituting the core of systems thinking as a distinguishable body of knowledge [18–21]. Approaches like soft systems methodology (SSM) [22], interactive planning [23] or critical systems heuristics (CSH) [24] use both systems models and systems-based inquiry activities to help people question the values that inform decisions and explore—through dialogue and debate—future possibilities for action to improve a perceived situation.

Checkland [22] conceives of an enquiry as a system whose main goal is to promote *learning* about the situation at hand by its different stakeholders; in such an enquiry, which he names as soft systems

methodology or SSM, -systems techniques and models can be used to enable stakeholders to explore connections between their perceived problematic issues and possibilities for improvement. Ackoff's interactive planning approach [23], or IP, brings the idea of a mess, a set of interconnected problems which, together, conceived of as a system, can be tackled by continuously enquiring about and acting on possibilities to improve quality of life as a whole.

Based on Kant's use of practical reason, Werner Ulrich's critical systems heuristics (CSH) [24] approach provides a series of critical questions that represent "whole" knowledge categories, whose collective discussion and debate could help situation stakeholders generate defensible and sustainable conceptions and decisions for present and future generations. In addition, Midgley [18]'s extension of Churchman's idea of a systems boundary considers wider societal situations of marginalization, in which people and issues (represented as boundaries) take the status of sacred/profane in the analysis of and decision making about situations. Marginalization can also be manifested when experts or decision makers select approaches or methodologies to facilitate enquiry [2]. All these types of boundaries could be surfaced and debated upon in relation to their ethical impacts for various stakeholder groups and using the aforementioned systems approaches (or parts of them).

Other systems thinkers like Jackson and Keys [25], Flood and Jackson [26], Flood and Romm [27], Jackson [19] and Reynolds and Holwell [21] also provide extended categorizations to assess the relevance of the above systems approaches in relation to problem situations where they are used to support enquiry. These situations or contexts can be characterized by a singularity or plurality of issues to tackle, as well as who can be considered a relevant participant [25]. Building on this characterization, meta-methodological frameworks like total systems intervention (TSI) suggest that by using metaphors to relate to situations, facilitators and participants can creatively flesh out issues that require attention and select the most relevant systems approaches to tackle them [26,27].

Internally, though and between systems thinking communities, there is a lack of consensus between systems thinkers about the value of continuing to foster paradigm-based research and practice [21,28], in particular regarding how systems approaches deal with issues of power [21,29] and how some communities which involve people from academic or practical circles could find value in theoretically informed use of systems approaches in the light of emerging (i.e., post-normal, post-paradigmatic) phenomena and situations [21].

To address the above tensions, further developments in systems thinking establish a distinction between systems approaches, methodologies and methods [18,21,27]. These developments also highlight how systems thinking as a whole body of knowledge has recognized a common transition of communities from gaining "holistic" understandings of situations (i.e., via approaches or methodologies) to then proposing ways of supporting their diversity and continuous adaptation [21]). A key element enabling this common grounding is the recognition of people and our creativity in the activity of the structuring of problems and the design of enquiry approaches [12,13,21,29–32].

Manifested in the emerging identification of "uncertain messes" like the abovementioned one, the coronavirus world pandemic offers systems thinking the opportunity to continue, if not enhance, inter-(disciplinary, professional) work, a type of work that is now fueled by the use of technology-mediated forms of interaction [4,33]. We could promote our critical knowledge and awareness to audiences with approaches that allow for the participative surfacing and exploration of possibilities for improvement in complex (post-)pandemic situations. However, the ownership of these situations is somewhat in disarray, up for grabs and at risk of reinforcing situations of marginalization [33,34]. As systems thinkers, we need to be both open to, as well as critical of, this.

In the 1990s, Corbett and Van Wassenhove [35] reignited debates about the future of OR as a profession. These authors attributed a perceived lack of OR relevance of to a "natural drift", an evolutional stage in which the ability of OR to help tackle messy situations needed to be revisited in light of changing circumstances that included the emergence of different professions and disciplines in science. An emerging role for OR experts as *brokers of knowledge* between academia and practice,

between theoretical and applied knowledge, was proposed by these authors to help mediate the interactions between users, theoretical or practical experts, decision makers and other stakeholders.

With this idea of knowledge brokering and considering a common transition between holism and pluralism, we could also follow less-established traditions within systems thinking, and conceive of ways to help us and others better understand the emerging dynamics of knowledge and their intertwining with ethics in society in (post-)pandemic situations. To meet this aim, in the following sections, two distinct but complementary scenarios to support future systems thinking work are offered.

## 3. A First Scenario: Inter-Work

This scenario for systems thinking sees possibilities to bring together different works in academia and elsewhere to facilitate participative knowledge enquiry into the complex, multi-faceted systemic nature of (post-)pandemic messy situations. This scenario might not be seen as much different from what systems thinkers have become distinguished for in its support of knowledge areas like community development, environmental management, sustainability education, information systems or disaster planning as mentioned before, and also using systems models [36]. However, it is now essential to consider that, perhaps differently from the times of Churchman and Ackoff, interest in speedy, data- and technology-mediated knowledge generation and brokering is currently fueling continuous competition and marginalization between groups [2,33].

The sociologist Andrew Abbott [5–8] has called into question the accepted assumption in societies that that knowledge about the social is to be advanced, progressed or discovered. To him, and challenging Kuhnian ideas about how discoveries in science take place (i.e., mainly away from or bearing little relation to each other), this is an illusion. Key findings and differences about the social (i.e., positivism and interpretivism, narrative and causal analysis, realism and constructionism/constructivism, contextualism and non-contextualism, agency and structure, choice and constraint, conflict and consensus) have already taken place [7]. With this assertion, the so-called discovery of knowledge becomes its continuous fragmentation and specialization, which rather contributes to socially maintain "basket" systems or structures of academic disciplines and practical professions/para-professions that provide their participants with legitimate work opportunities in their societies [5,6].

Abbott [6,7] thus proposes considering knowledge in the social sciences as in a continuous and cyclical process of creative *rediscovery*. This means that the work of academic and practitioner groups is connected and mutually dependent [5,8]. Knowledge could be conceived of as contained in a "maze" [6]. Its unfolding comes and goes through different groups belonging to certain institutional or organizational locations and via continuous stages of differentiation, competition and absorption.

This means that groups compete over control or ownership over specific knowledge problems (diagnoses) and prescriptions or treatments for them, as well as inferences (classifications or taxonomies between problems and treatments [5]. Through time, groups maintain, expand or lose (vacate) such ownership, the knowledge and work that it entails. "New" knowledge distinctions emerge through these stages to replace "old" ones but also to absorb their claims (about what is a problem/phenomenon and how to study/deal with it), thus ensuring that valuable knowledge and the social relations that it embeds is being kept alive in societies. The cycles of rediscovery can be portrayed as shown in the Figure 1 [6,12].

The cycles of knowledge rediscovery of Figure 1 could help us make sense of how the coronavirus (post-)pandemic is being tackled and what systems thinking can do about it. Firstly, some important connections between different groups in academia, government or industry have surfaced or have been fostered. For instance, scientific articles published by prestigious journals (e.g., The Lancet®) or new vaccine trials have quickly impacted government policies worldwide. This has also revealed a strong and continuous degree of competition and connectivity between groups and their physical/institutional locations to assemble, complete and disseminate knowledge in our societies [6,8,18,33]. In the name of

"good work", some of these connections might include (but are not limited to) professional or funding relationships between academic departments, associations and industry [7,37].

Secondly, what Abbott also proposes with Figure 1 is that because of situations like the world pandemic, cycles of knowledge (re)discovery could be *shortened* by creative groups entering into them by borrowing knowledge from others. More novel or quirky ideas could be obtained from quickly analyzing electronic data or using simpler knowledge or technologies. With this, competition and marginalization could also be increased, upon which systems thinkers need to exert appropriate knowledge brokering. And thirdly, as per Abbott [5–8], systems thinkers would also need to raise awareness about the mutual if not systemic dependence between "winners" and "losers" of the (post-) pandemic dynamics of knowledge [2,33], given that social relations among them are *still needed* to surface, maintain or use key knowledge concepts, approaches or ideas to benefit and provide diverse audiences in society with legitimate work opportunities.

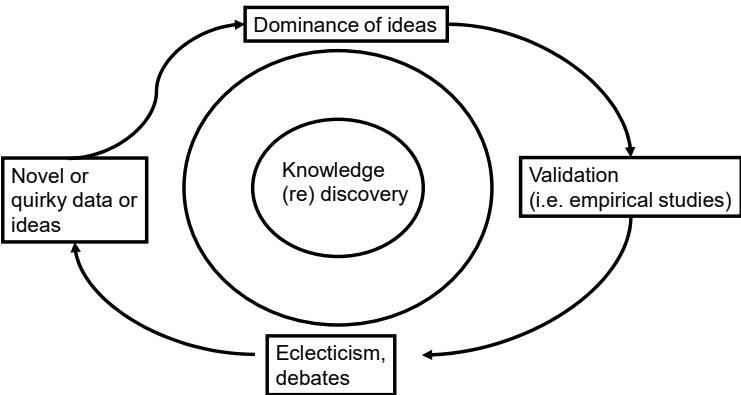

**Figure 1.** Cycles of knowledge rediscovery.

It follows that a future (post-)pandemic scenario of inter-work could see systems thinkers becoming interested, engaged or taking part in inter-disciplinary initiatives. In those, we could invite others to adopt a perspective of how knowledge could be rediscovered if not reconfigured in simpler ways and within the relations that have made it "legitimate". The world pandemic has highlighted for instance how going back to healthy living habits (eating, cycling, exercising) could be a positive contributor for stress or anxiety reduction. This knowledge as well as how it could be quickly validated, disseminated and adopted could be revisited, brokered and supported with the help of systems approaches in order to visualize social relations and processes that (could) contribute to its inclusive (re)discovery. Systems thinkers could consider the following strategies:

- Finding and filling knowledge gaps by providing new classifications (diagnoses, treatments, inferences) and thus promoting eclecticism in the above cycle, as well as filling "vacated" activities of instruction, problem solving or research. For instance, during the world pandemic vaccine generation might have left a space for vaccine prevention; social distancing could be seen as leaving a gap for social intimation. Groups in social science could be invited to work together to fill these or other gaps.

- Borrowing and simplifying distinctions from others (diagnoses, treatments and inferences) and promoting eclecticism, as well as quirky and novel ideas, that are more generic and portable [5–7], so that they become better understood and adopted by diverse audiences. This often would mean translating potential competitors' claims into one's own terms by questioning their validity or applicability [6]. To the abovementioned returning to normal (post-)pandemic situations via distancing, there might be psychological or sociological concepts whose translation in systems thinking terms could be simpler to understand by different groups (i.e., citizens, communities, local governments) [2].

### 3.1. Implications

With the above possibilities, an inter-work scenario could also enhance existing opportunities to conceive of problematic (post-)pandemic situations. Systems thinking researchers and practitioners might find that they can extend the scope of their creativity. By borrowing knowledge distinctions (i.e., diagnoses, treatments, inferences, others) and from different disciplines, professions or other groups, people could also explore their lineages and venture to suggest other, adjacent possibilities. They could ask questions about "what if" problems that are conceived of and tackled in alternative ways [7,12].

For instance, the study of lockdown situations using qualitative methods could lead researchers to conceive of how such methods had been enriched with quantitative techniques (eclecticism, quirkiness when rediscovering knowledge). They could also bring distinctions about agency and structure [7], in order to ascertain how both of these bring different routes of conceiving of and acting about lockdown situations.

Despite describing how knowledge unfolds in the face of inter-group competition, and how to deal with resulting knowledge marginalization situations via translating and making knowledge more portable, Abbott [6] advocates an overall collective sense of *solidarity and understanding* between individuals and groups working in the social sciences, in which they could engage in fruitful and mutual dialogue with others to learn where they have been on their knowledge (re)discovery journeys, and in doing so they can recognize their own creative limits and opportunities when borrowing knowledge from others.

As knowledge brokers, this first scenario of inter-work could invite us all to become (more) reflective, inclusive and less driven by a sense of urgency, and thus to maintain a socially oriented and healthy balance of thinking and action. This might sound counter intuitive, given the sense of opportunity and urgency that is being identified for systems thinking [2–4,21]. We could continue exploring knowledge brokering opportunities whilst critically challenging established knowledge traditions or lineages (i.e., quantitative, qualitative) that might be contributing to the generation of unintended marginalization effects in (post-)pandemic populations or groups. We could be more explicitly guided by an attitude of self-observation, understanding, practical relevance, solidarity and community understanding [6,12,21].

### 3.2. An Example

Mental health has become a very important societal issue in many countries. As private and non-private health providers struggle to cater for an increasing numbers of mental health users, some opportunities are being offered by the incorporation of information and communication technologies (ICTs) to support activities of online therapy and counselling, among others.

With the pandemic, however, there is evidence of an increase in mental health issues in adults. Adopting a systemic perspective of this complex situation could bring researchers or practitioners from different disciplines (psychology, human computer interaction, social work) to work together; there could be opportunities to tap into government funding that would need to be evaluated carefully so as not to continue reinforcing habits or practices that could exacerbate anxiety or isolation.

A systems thinker entering into inter-professional work opportunities could invite others to consider *where* the issue of ICT-mediated mental health support has previously been researched or discussed, and with the aim of enhancing context sensitivity, awareness or user participation. She could also invite them to identify vacated areas (i.e., non-behavioral issues related to the situation), and guide reflection and discussion for knowledge rediscovery with questions like:

✓ What knowledge could we borrow from others? Where have they been, or gone with it? For which purposes? What limitations and opportunities have they encountered?

✓ What if this situation is creatively considered as a one of agency versus structure, political versus economic discourses, determined versus socially constructed habits, or could be researched as

interpretive versus positivist, of a quantitative versus qualitative nature, or as one of choice versus constraint?

✓ What concepts and approaches from systems thinking or participating disciplines, professions or others could help us strengthen our social relations to support better knowledge rediscovery, dissemination and use in society?

✓ How can we maintain a sense of solidarity, mutual understanding and compassion?

As initial answers to the above questions, systems thinkers could highlight the value of conceiving the situation as not only technical but human, mapping connections between groups contributing to knowledge about mental health during the pandemic and getting to know their research or practice "routes" or pathways. They could also borrow knowledge from alternative medicine or therapies or well-being (healthy eating or better sleep, seeking and receiving face-to-face emotional support) to fill identified gaps in knowledge, and in doing so they could facilitate the improvement of communication opportunities between health and non-health professionals, decision makers and other stakeholders.

In promoting the above, systems thinkers could use the aforementioned approaches like soft systems methodology (SSM) or interactive planning (IP) to map connections between knowledge/groups who are and could contribute to study and act on the connections between issues of mental health, ICT and the pandemic. They could also use systems dynamics and complexity models [36], as well as critical approaches like CSH to invite others to problematize and rediscover social relations, ways of interacting or communicating which, as collectively built leverage points, could help us better understand and address manifestations and consequences of (post-)pandemic situations (i.e., self-isolation, physical distancing).

## 4. A Second Scenario: Ethical Vigilance

With his proposal for cyclical knowledge rediscovery across groups and work locations, Abbott [6] leaves it to individuals to "do the right thing" about (not) joining, supporting or connecting groups or (not) borrowing their knowledge claims and their underlying social relations. In Abbott's view, it seems as if it is ultimately society which somehow self-regulates and "picks" what knowledge remains valuable.

As individuals, we now know that the coronavirus pandemic and actions to deal with it have physical as well as emotional impacts on ourselves and our communities, and we have often felt confused if not passive or powerless to challenge the "evidence" that supports governing thinking and action [33,34]. What can we do about this?

The holism that could be gained by inter-work also needs critical (self-) awareness and action within power relations towards a critical and participative transition towards plurality. Knowledge brokering would need to be supported by an ethical and (inter-)individual dimension of thinking and practice. This is to ensure that (post-)pandemic knowledge in its various forms does not become the new "normal" or dominant "traditions", which could thus constrain the flourishing of "other" possibilities for human creativity to deal with (post-)pandemic situations.

Foucault [9] (pages 102–103, italics added) coins the term governmentality to account for:

1. The *ensemble* formed by the institutions, procedures, analyses and reflections, the calculations and tactics that allow the exercise of this very specific albeit complex form of power, which has as its target population, as its principal form of knowledge political economy, and as its essential technical means apparatuses of security.

2. The tendency which, over a long period and throughout the West, has steadily led towards the pre-eminence over all other forms (sovereignty, discipline, etc.) of this type of power which may be termed government, resulting, on the one hand, in the formation of a whole series of specific governmental apparatuses, and, on the other, in the development of a whole complex of *savoirs*.

3. The process or rather the result of the process, through which the state of justice in the Middle Ages, transformed into the administrative state during the fifteenth and sixteenth centuries, gradually becomes 'governmentalized'.

With the above, Foucault proposes the structuring of collective or individual *conduct* of people within our social relationships (including those with ourselves) of power. Relationships manifest themselves in the continuous inviting, seducing, nudging, rejecting, challenging or even creating individually or collectively forms of thinking and acting [38]. The sense of immediacy, uncertainty and urgency that seems to prevail in the pandemic is to some a manifestation of forms of governmentality which are not imposed but structured with collective imaginaries, like solidarity, saving lives or helping out the economy [33,34,39].

What Foucault's governmentality brings to the knowledge "maze" of rediscovery proposed by Abbott [6] is a possibility of "reimagining it whilst being in it". A second scenario for systems thinking could provide us with a sense of "ethical vigilance" of such a maze, in order to identify which could be the main dangers to ourselves as free, (self-)governable subjects [38] when we embark on (post-)pandemic knowledge rediscoveries. To the available or resulting forms of regulating conduct during relevant situations (i.e., lockdown, social distancing, use of masks, staying at home, working from home), our task is that of helping identify the systemic conditions that led to their (re-)emergence and adoption [40,41]. We would need to provide different understandings of how such "events" and practices have arisen; how they could be privileging certain types of thinking and acting for ourselves and others in dealing with events; how they could be marginalizing "other" forms of subjectivity [13,18,33,36]; and what we could be doing about it.

In this second scenario, we could use systems approaches and ideas to help ourselves and others holistically visualize the unfolding of (post-)pandemic-related knowledge (including that emerging from inter-work), as well as its impacts on both individuals and populations, conceiving of them as historically and contingently formed to regulate human conduct. To those, we could bring forth "other" forms of being, promoting the individual or collective adoption of "old" and new "selves", with their habits or practices [13]. Additionally, by doing this, we could decide what is no longer necessary for the "new normal" constitution of ourselves as individual or collective subjects of both knowledge and ethics [33,42].

Alongside systems approaches or other forms of enquiry to help ourselves and others reflect on new forms of relation (i.e., meditation, writing, play, humoring), some questions that could help develop the second scenario of ethical vigilance are:

1. How did we come to be who we are, even when trying to deal with pandemic situations or being conducted to do so?
2. What (un)necessary constraints are we to live with in a new, idealized new normal for selves [16]?
3. What practices from our "old" or "new" selves could we (re)discover to help us comply with what is required by the new normal whilst giving us renewed senses of ethical purpose about it?
4. What else or who else could we include in new relations of self–others?

*4.1. Implications*

This second scenario could invite us to assume fewer formal roles or practices, inviting us to use our imagination, passions or interests to fuel new normal journeys, individually and collectively. Its adoption would mean working less at the level of knowledge disciplines/groups and more at the intra-, inter-individual ones of relationships; this is something that has already been recognized in the systems thinking field [21], but would require further and critical thought when it comes to power relations that impact the senses of "self". As systems thinkers, we could even become brokers not only of knowledge but of whole ways of being: Less focused on the pursuit of systemic, practical or emancipatory knowledge, and more on strengthening locally feasible, meaningful relations or connections between ourselves and others.

Alongside knowledge rediscovery and the pursuit of mutual understanding, solidarity and compassion (first scenario), with this second scenario, we could become more (self-)compassionate: there is no need to frantically run or race to the top anymore (what many "old" selves including this author strived to do pre-pandemic), but to find new meaning in who we are or can be.

*4.2. An Example*

It has been widely argued that creativity is a key skill or competence to be nurtured in management students worldwide [43]. Under the name of "innovation", education has been institutionalizing the practice of "innovation challenges" to encourage students to become more entrepreneurial and self-confident, so that they can also contribute to innovation in large commercial organizations and technology providers. In the main, innovation challenges require educators and students to simulate "real" environments where they compete to generate feasible and profitable solutions.

Furthermore and with the world pandemic, information and communication technologies, or ICTs, have contributed to make work and educational practices more remote than used to be the case. Assessment rubrics have been promoted as a way of standardizing online course evaluation of creativity [44]. With this, there is the risk of normalizing practices so that they become self-governing technologies [45], meaning that these practices establish their own norms and ways of complying with them that do not have much to do anymore with regulating human conduct for the benefit of society as a whole. For instance, the excessive and ICT-mediated surveillance of social distancing or educational attendance could become a system of its own which, in the long term, could exacerbate rather than reduce conflict.

To the above situation, an educator adopted the proposed second scenario to explore *how is it that we have all become governed due to the pandemic*. She identified historical and contingently formed links between innovation and economic growth, and this led her to realize that there might be a need to keep her desire to educate somehow separate from governing strategies (achieving employability of students). She also accepted, albeit not without difficulties, "online assessment failure" in education, given that, no matter what, students and institutions will never be 100% happy with it. She decided to keep her own well-being in balance with work and other "selves" [46].

The educator used an existing course on process management to bring one of her old selves (a cyclist and a recycler) to be revealed as a way of telling her students that it is important to preserve relationships with "what/who makes you tick" in life. She also got in touch with one bike recycling organization and organized a student visit. From the visit, the educator set up an "innovation challenge" for students to redesign a process to help this organization better fulfil its educational *and* service functions to society, whilst giving students a sense of purpose and local benefit.

Together with "dominant" knowledge on process management (Six Sigma, lean), creativity techniques and systems approaches (SSM) were used to guide students' learning in this course. Technology-mediated content or interactions (i.e., an online rubric to mark the assessment) were also developed just after lockdown to help students continue with their learning and work together to meet stated learning outcomes [44]. Whilst this initiative could also be seen as a form of knowledge rediscovery, with the educator acting as a broker between university and social enterprise organizations, a focus on ethical vigilance helped the educator to become more aware that her old selves (as a teenager she liked to cycle with her friends from her school and neighborhood) need(ed) to be brought forth again. She initially adopted this old self by buying an electric bike to cycle to work, realizing soon that she was not a young enthusiast anymore. She accepted failure again in order to make peace with her old and new selves (one of which cares for her well-being).

This also means that, as a "whole", she no longer follows a sense of urgency or perfection in what she does or thinks. She is learning to trust her different selves as well as the local bike repair shop. In the new normal, she keeps her support relationships as a mentally fragile, dedicated and caring self and hopes that she can adequately share her own anxieties with her students and others and within existing "new normal" constraints.

## 5. Conclusions

In this paper, it has been argued that during and after an "unprecedented" situation like the world coronavirus pandemic, systems thinking has started offering possibilities to contribute to societal improvements. Systems thinkers could continue acting as valuable brokers of knowledge. To do so, however, a more in-depth and reflective understanding of the current dynamics of knowledge in societies and their intertwining with ethics is needed.

To meet this need, this paper has proposed two different but complementary scenarios to support future systems thinking and action during or after the world coronavirus pandemic. The paper has offered detailed features, guiding questions, implications and examples for the adoption of the scenarios proposed.

Each of the proposed scenarios also brings commitments to systems thinkers and those who we could help in (post-)pandemic situations. An emphasis on knowledge rediscovery embeds a commitment to solidarity, mutual understanding and dialogue between groups concerned with or affected by knowledge production during and after the world pandemic of coronavirus. An emphasis on ethical vigilance invites us all to be aware of risks and opportunities for ourselves and others that (could) (re-)emerge for ourselves, other people and the planet, and act creatively to address them.

The current climate of uncertainty that many of us live in could be seen as an opportunity to stop and reflect on how the 'new normal' needs to change; perhaps similarly, as well as differently, from Ackoff, Churchman or other systems thinkers and communities, the future of systems thinking could reside not only in solving the "right" (complex) problems or in appropriately brokering knowledge, but also in assuming the "right" attitudes towards the current and future (post-)pandemic situations we are part of. With compassion as a renewed value, the proposed scenarios bring to light the importance of promoting inclusivity and critical reflection about how we have or could contribute to discovering valuable knowledge, and how we could manage its diverse systemic impacts for ourselves and others.

The hope is that the insights of this paper could help us stop, pause and reflect on what type of selves, organizations, societies and universe we want to be and live in in the new "normal", and how we could move, slowly but surely, together, critically but also compassionately, towards our desired ways of life in academia, practice or life in general.

**Funding:** This research received no external funding.

**Acknowledgments:** The author would like to thank his family for the emotional support received during the writing of this paper, and the editors and reviewers for their valuable comments and suggestions to improve its content.

**Conflicts of Interest:** The authors declare no conflict of interest.

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
