# Peer review of "Inter-Work and Ethical Vigilance: Two Scenarios for the (Post-)Pandemic Future of Systems Thinking"

_systems, doi:10.3390/systems8040036_

Round 1
Reviewer 1 Report
This paper is well-written albeit lacking of its scientific rigour. In my opinion, this article seems more like a short communication paper rather than a full article. Scenarios and discussions does not derive from primary data. To generate valid and unbiased claims, I suggests this paper to use an inclusive and participatory research design.
Furthermore, the ideas presented in the first scenario is not new. Working with stakeholders across disciplines have been discussed really well in the participatory systems thinking method. In my opinion, the future of AST post pandemic is digitalisation for inclusivity of geographically spread stakeholders.
Author Response
This paper is well-written albeit lacking of its scientific rigour. In my opinion, this article seems more like a short communication paper rather than a full article. Scenarios and discussions does not derive from primary data. To generate valid and unbiased claims, I suggests this paper to use an inclusive and participatory research design.
RESPONSE: Many thanks for your thoughtful review. Yes, this is a conceptual paper that looks at the possible unfolding of systems thinking as a body of knowledge using theoretical ideas to inform future research and practice in this body of knowledge. The scenarios proposed have been further detailed in their features, implications and examples to offer the reader a clearer view of their nature.
Furthermore, the ideas presented in the first scenario is not new. Working with stakeholders across disciplines have been discussed really well in the participatory systems thinking method. In my opinion, the future of AST post pandemic is digitalisation for inclusivity of geographically spread stakeholders.
RESPONSE: Thank you for this suggestion. For the first scenario, its detail has been enhanced and related to inter-work (disciplinary, professional) so as to differentiate it from current systems thinking and practice. The questions and example of this scenario and of the second scenario have been amended to reflect also possibilities for digitally mediated interaction.
RESPONSE: Many thanks for your thoughtful review. Yes, this is a conceptual paper that looks at the possible unfolding of systems thinking as a body of knowledge.
Reviewer 2 Report
Firstly, I invite you to use ‘approaches’ rather than ‘methodologies’; the former being more generic set of methods associated with particular traditions, and the latter being precise, very context specific, application of methods (notwithstanding Checkland’s use of the term in SSM).
I think the crux of the paper is good and generally well structured in relation to the 5 sections. The core two ideas are I think relevant and worthy of attention (I note below some possible alternative re-phrasings of these two ideas which may help with the narrative, but nothing that distracts from the substantive ideas presented). My main concern is in relation to the positioning of ideas. I think the paper can be significantly enhanced by making reference to the wider set of traditions on which it draws. Whilst possibly taking away the ‘innovative’ stance being presented, an improved contextual positioning would in my view invite greater resonance amongst readers who are familiar with these traditions, and/or lessen the risk of the content being regarded as simply recycled ideas presented in different clothing (which is ironic, given reference to Abbott’s observations on ‘knowledge rediscovery’ to which the paper draws).
Section 1: I have a problem with opening section here: “Applied systems thinking (AST) is a set of systems ideas, concepts and methodologies for problem solving in organisations. Under different names (i.e. soft operational research, complexity, systems thinking) and after a few decades, it is now a recognised body of knowledge that serves professions or disciplines including operational research (OR) and management, both in research and practice (Mingers and White, 2010).” (my italics)
I’m not familiar with the Mingers and White reference, but I am familiar with traditions of systems thinking in practice. Aside from a (rather obscure) journal named AST (which is not referenced in any of the citations), I am not familiar with the term Applied Systems Thinking. Giving it such prominence (with supposed ‘recognition’) and an associated acronym is rather deceptive in my view. Given it’s lack of currency up until now, if using the phrase I think it better for you to take some ownership for it’s use. You can circumscribe this ownership in relation to the purpose of the paper being written.
(For an outline discussion on traditions informing systems thinking in practice see Ramage and Shipp (2020) Systems thinkers (2nd edition) and/or Reynolds, M. and Holwell, S. (2020) Systems approaches to making change: a practical guide with Preface to 2nd edition, and particularly chapter 1 Introducing systems approaches. In: Martin Reynolds and Sue Holwell eds. Systems approaches to making change: a practical guide. 2nd Edn. London: Open University and Springer, pp. 1–23.)
You’ll find that systems thinking in practice (STiP.. a name adopted by The Open University for it’s postgraduate suite of qualifications since 2010), informs and supports disciplines far wider than OR and management studies, including health practice, policy studies, and engineering amongst many others relevant to pandemic)
Contemporary STiP ideas (from mid 20th century) being drawn upon in this paper, deals not with ‘problem solving’ (systematic) so much as ‘problem structuring’ (systemic), which is significantly different. Of course, good STiP involves the interplay between systemic and systematic.
Section 2: I’m understanding the division between x2 scenarios in terms of ‘doing’ AST, and ‘being’ a systems thinking practitioner (STP). (Incidentally you may like to know the ASTiP group at OU have successfully steered UK government approval of, and a recognised professional Standard for, a postgraduate apprenticeship for STP in 2020…So it’s official, there is a recognised job description of a systems thinking practitioner, at least in UK!). I think some consistency with these terms – doing and being - might help the narrative, which can at times be a little wayward. (Note for example, in Conclusion you comment on ‘being and thinking’ …)
I wonder how much the distinction you make might tally with Habermas’ distinction between System and lifeworld. But then it seems more akin to a divide between ‘collective’ (solidarity) and the ‘individual’. Either way, I think some clearer explanation and (ironically recycled…) roots of the distinctions that you are making need to be expressed. The distinction also speaks to traditions of ‘communities of practice’ (Wenger, and scenario 1) and social learning (in both scenarios but perhaps more with scenario 2). In our own teaching in the past 10 years at ASTiP (applied systems thinking in practice group) on ‘making strategy with systems thinking in practice’ (OU module code: TB871) we use two parallel streams of pedagogy – a Tools stream (teaching 5 systems approaches) and a People stream (focusing on what it means to be systems thinking practitioner). So of course I see a lot of pleasing resonance with this paper.
I like the title ‘inter-work’ though not so impressed with ‘counter-conduct’ for scenario 2 (perhaps ‘counter-practice’ or possibly something that clearly joins the two, and helps with narrative like ‘inter-work practices’ and ‘counter-work practices’ (?)
Note “Midgley (2000)’s extension of Churchman’s boundary critique” …. I think you mean Ulrich’s idea of boundary critique. Churchman never used the term, though was clearly inspirational.
Good layout of sections 3 and 4 with general discussion followed by ‘implications’ and ‘an example’.
Section 3 (1st scenario): not sure how your use of ‘inter-work’ differs from ‘transdisciplinarity’ – a long-standing acknowledged feature of STiP. Though I do like your (?) term inter-work. Just needs more grounding in references to shift from interdisciplinarity to transdisciplinarity.
Although the work of Abbott is interesting (and new to myself), I’m not sure what it is adding to the already existing constructivist epistemology on which contemporary systems thinking is clearly established. My guess is (from titles of papers referenced to Abbott) is the specific dimension to professional silos and their links with disciplinary silos. Possibly make a link here to Wenger’s communities of practice(?). This I think is important, but needs flushing out more in order to make Abbott’s distinctive contribution over and above Kuhn more relevant. Perhaps a little acknowledgement that it comes from this constructivist tradition is all that’s required. The thing I like about what I’ve read is the notion of ‘rediscovery’ and the idea that knowledge goes through some kind of recycling (though the term is not actually used). I think also the idea that social science is comprised of siloed disciplinary ‘baskets’ is very helpful: “Through a so-called discovered, fragmentation and specialisation is contributing to maintain ‘basket’ systems or structures of social sciences academic disciplines and practical professions, and ultimately provide their participants with legitimation before societies.” But again, worth drawing on and explicating difference with Thomas Kuhn’s paradigms. Some reference to Kuhn’s work would help ground the reader in the epistemological arguments being made. Ideas of ‘competition’ and ‘connectivity’ are clearly relevant key drivers for Abbott in relation to professions and disciplines in pandemic age. Kuhn’s relevance is also significant to well established tradition of ‘post-normal science’ in established field of Science and Technology Studies (STS); clearly of importance in any discussion of ‘new normal’ arising post-pandemic (?).
Are there not pockets of ST practitioners that exhibit this competitiveness? Perhaps needs commenting on; particularly between hard and soft/critical systems thinkers (taking different epistemological stances on use of ‘systems’ idea – ontological vs nominal) but also, as you might be aware, amongst so called critical systems practitioners themselves (contingency approach of Jackson vs more universal - boundary critique - approach of Ulrich and Reynolds).
So how might AST help in nurturing a better more ‘cooperative’ form of ‘correspondence’ between practitioner communities, other than rather flaky calling for more reflection, inclusion, and compassion? What is missing for me is some more sense of what is in the practice of AST that might help engagement with different perspectives in order to foster the transdisciplinarity needed as suggested by writers like Abbott. The example of mental health intervention is a good one. It is an example of problem structuring (not problem solving…) using SSM. So I think from the outset you need to highlight problem-structuring methods as distinct from problem-solving methods as being something of particular value in AST. But also there is a need for some more elaboration on why these methods are described as AST, and (albeit summarily) how they are administered, so as to make them, in your view, systems thinking methods. Again, for me, it’s the interplay between systematic and systemic that is core to contemporary tradition of STiP or AST (as distinct, for example, from traditions of systems science and complexity science, where the focus is on systematic analysis)
I think some more play on practitioner adaptability to different contexts and changing circumstances might be made, possibly along lines of bricolage. So a systems thinking practitioner can draw on many different traditions of practice in order to make relevant the interventions being constructed.
“scientific articles published by prestigious journals (e.g. The Lancet® ) have quickly impacted government policies worldwide, only to be in some cases retracted and causing policy reversal.” …perhaps needs some reference for evidence.
Section 4 (2nd scenario): I’m a little uneasy with notion that this ‘scenario’ is innovative and creative, suggesting by implication that scenario 1 is perhaps not (?). There is here though good emphasis on ethics (and I would suggest, perhaps more importantly, politics – with small ‘p’ – i.e. relations of power) counter-knowledge and possible need to preserve knowledges that are at risk of being further marginalised or even destroyed. Promoting practices like meditating, writing (journaling?), reflecting, and self-isolating, can appear to be a little fluffy without providing the traditions of (systems) practice on which they are, or can be, built. Here I’m thinking of Otto Sharmer’s work on presencing and Theory U, and traditions of ‘social learning’ (Chris Blackmore and Ray Ison). I wonder also if you might bring in the importance of marginalisation in AST as expressed through Churchman’s seminal (1979) publication The systems approach and its enemies’ (curiously not referenced), and of course the extension of this by Ulrich with CSH (see latest edition of this… in Ulrich, W. and Reynolds, M. (2020). Ch. 6. Critical Systems Heuristics: The Idea and Practice of Boundary Critique.)
I like the idea of playful vigilance, but again wonder if this might also be applied to both scenarios?
You may like to read Jon Harle blog posting cracks in the knowledge system: whose knowledge is valued in a pandemic and beyond? 28/08/2020. Which speaks to this paper rather nicely in my view.
Author Response
Firstly, I invite you to use ‘approaches’ rather than ‘methodologies’; the former being more generic set of methods associated with particular traditions, and the latter being precise, very context specific, application of methods (notwithstanding Checkland’s use of the term in SSM).
RESPONSE: Many thanks for the detailed comments and suggestions. I have used the word 'approach' throughout the text, and also brought to the fore some current debates in the domain of systems thinking.
I think the crux of the paper is good and generally well structured in relation to the 5 sections. The core two ideas are I think relevant and worthy of attention (I note below some possible alternative re-phrasings of these two ideas which may help with the narrative, but nothing that distracts from the substantive ideas presented). My main concern is in relation to the positioning of ideas. I think the paper can be significantly enhanced by making reference to the wider set of traditions on which it draws. Whilst possibly taking away the ‘innovative’ stance being presented, an improved contextual positioning would in my view invite greater resonance amongst readers who are familiar with these traditions, and/or lessen the risk of the content being regarded as simply recycled ideas presented in different clothing (which is ironic, given reference to Abbott’s observations on ‘knowledge rediscovery’ to which the paper draws).
RESPONSE: Thank you for the suggestion. The paper has been revised throughout and its positioning amended. Current debates in the systems thinking domain have been highlighted and the possible pathways identified. It has been argued that the coronavirus pandemic has impacted on the configuration of social science knowledge and thus there are opportunities for systems thinking (i.e. inter-work), only that they need to be taken with a degree of openness and critical caution. An alternative pathway for the future(s) of systems thinking has been proposed and developed using Abbott and Foucault's ideas.
Section 1: I have a problem with opening section here: “Applied systems thinking (AST) is a set of systems ideas, concepts and methodologies for problem solving in organisations. Under different names (i.e. soft operational research, complexity, systems thinking) and after a few decades, it is now a recognised body of knowledge that serves professions or disciplines including operational research (OR) and management, both in research and practice (Mingers and White, 2010).” (my italics)
I’m not familiar with the Mingers and White reference, but I am familiar with traditions of systems thinking in practice. Aside from a (rather obscure) journal named AST (which is not referenced in any of the citations), I am not familiar with the term Applied Systems Thinking. Giving it such prominence (with supposed ‘recognition’) and an associated acronym is rather deceptive in my view. Given it’s lack of currency up until now, if using the phrase I think it better for you to take some ownership for it’s use. You can circumscribe this ownership in relation to the purpose of the paper being written.
RESPONSE: Thank you for this insight. The introduction of the paper has been reworded to account for systems thinking (not only applied systems thinking). The AST acronym has been replaced by 'systems thinking'. And the inclusion of practice has been also highlighted throughout the whole paper.
(For an outline discussion on traditions informing systems thinking in practice see Ramage and Shipp (2020) Systems thinkers (2nd edition) and/or Reynolds, M. and Holwell, S. (2020) Systems approaches to making change: a practical guide with Preface to 2nd edition, and particularly chapter 1 Introducing systems approaches. In: Martin Reynolds and Sue Holwell eds. Systems approaches to making change: a practical guide. 2nd Edn. London: Open University and Springer, pp. 1–23.)
RESPONSE: Thank you for this very valuable reference of Reynolds and Holwell. It has been used throughout the article to enrich the discussion and positioning of current debates between systems thinking communities, as well as highlight the transition of different systems thinking communities (holism, pluralism), also to argue for the importance of people when developing the future scenarios that are proposed in the paper.
You’ll find that systems thinking in practice (STiP.. a name adopted by The Open University for it’s postgraduate suite of qualifications since 2010), informs and supports disciplines far wider than OR and management studies, including health practice, policy studies, and engineering amongst many others relevant to pandemic)
Contemporary STiP ideas (from mid 20th century) being drawn upon in this paper, deals not with ‘problem solving’ (systematic) so much as ‘problem structuring’ (systemic), which is significantly different. Of course, good STiP involves the interplay between systemic and systematic.
RESPONSE: Thank you for this insight, it was considered to enhance the initial description of the different areas that systems thinking has contributed in the last few decades, as well as the aforementioned transition (holism, pluralism) both in research and practice which has been identified as a commonality between systems thinking communities.
Section 2: I’m understanding the division between x2 scenarios in terms of ‘doing’ AST, and ‘being’ a systems thinking practitioner (STP). (Incidentally you may like to know the ASTiP group at OU have successfully steered UK government approval of, and a recognised professional Standard for, a postgraduate apprenticeship for STP in 2020…So it’s official, there is a recognised job description of a systems thinking practitioner, at least in UK!). I think some consistency with these terms – doing and being - might help the narrative, which can at times be a little wayward. (Note for example, in Conclusion you comment on ‘being and thinking’ …)
RESPONSE: The ideas of Abbott were revisited and enhanced in the article to refer (not explicitly though) to the legitimation of work via connections between knowledge groups within and outside academia.
I wonder how much the distinction you make might tally with Habermas’ distinction between System and lifeworld. But then it seems more akin to a divide between ‘collective’ (solidarity) and the ‘individual’. Either way, I think some clearer explanation and (ironically recycled…) roots of the distinctions that you are making need to be expressed. The distinction also speaks to traditions of ‘communities of practice’ (Wenger, and scenario 1) and social learning (in both scenarios but perhaps more with scenario 2). In our own teaching in the past 10 years at ASTiP (applied systems thinking in practice group) on ‘making strategy with systems thinking in practice’ (OU module code: TB871) we use two parallel streams of pedagogy – a Tools stream (teaching 5 systems approaches) and a People stream (focusing on what it means to be systems thinking practitioner). So of course I see a lot of pleasing resonance with this paper.
RESPONSE: Thank you for this insight. As mentioned to other reviewers of this paper, the scenarios were described in more detailed. The ideas of Abbott were contextualised in relation to his challenge to paradigmatic thinking. Competition was highlighted as a key element of knowledge unfolding in society, but also the invitation of Abbott to promote solidarity, mutual understanding and compassion. In the first scenario, questions about where 'others' producing knowledge have been in their explorations were rewritten. As mentioned earlier, the consideration of people were also highlighted as a commonality between systems thinking communities from which the paper could stem and propose ways of working individually or collectively in the future.
I like the title ‘inter-work’ though not so impressed with ‘counter-conduct’ for scenario 2 (perhaps ‘counter-practice’ or possibly something that clearly joins the two, and helps with narrative like ‘inter-work practices’ and ‘counter-work practices’ (?)
RESPONSE: Thank you for this insight and the suggestion to employ better terms. The term 'counter-conduct' was replaced by 'ethical vigilance' throughout the revised article.
Note “Midgley (2000)’s extension of Churchman’s boundary critique” …. I think you mean Ulrich’s idea of boundary critique. Churchman never used the term, though was clearly inspirational.
RESPONSE: Thank you. The sentence has been rewritten.
Good layout of sections 3 and 4 with general discussion followed by ‘implications’ and ‘an example’.
RESPONSE: Thank you. This structure has been maintained.
Section 3 (1st scenario): not sure how your use of ‘inter-work’ differs from ‘transdisciplinarity’ – a long-standing acknowledged feature of STiP. Though I do like your (?) term inter-work. Just needs more grounding in references to shift from interdisciplinarity to transdisciplinarity.
Although the work of Abbott is interesting (and new to myself), I’m not sure what it is adding to the already existing constructivist epistemology on which contemporary systems thinking is clearly established. My guess is (from titles of papers referenced to Abbott) is the specific dimension to professional silos and their links with disciplinary silos. Possibly make a link here to Wenger’s communities of practice(?). This I think is important, but needs flushing out more in order to make Abbott’s distinctive contribution over and above Kuhn more relevant. Perhaps a little acknowledgement that it comes from this constructivist tradition is all that’s required. The thing I like about what I’ve read is the notion of ‘rediscovery’ and the idea that knowledge goes through some kind of recycling (though the term is not actually used). I think also the idea that social science is comprised of siloed disciplinary ‘baskets’ is very helpful: “Through a so-called discovered, fragmentation and specialisation is contributing to maintain ‘basket’ systems or structures of social sciences academic disciplines and practical professions, and ultimately provide their participants with legitimation before societies.” But again, worth drawing on and explicating difference with Thomas Kuhn’s paradigms. Some reference to Kuhn’s work would help ground the reader in the epistemological arguments being made. Ideas of ‘competition’ and ‘connectivity’ are clearly relevant key drivers for Abbott in relation to professions and disciplines in pandemic age. Kuhn’s relevance is also significant to well established tradition of ‘post-normal science’ in established field of Science and Technology Studies (STS); clearly of importance in any discussion of ‘new normal’ arising post-pandemic (?).
RESPONSE: Thank you for this in-depth insight. As mentioned earlier, the ideas of Abbott have been revisited and related as challenging Kuhn and current systems thinking pathways. The term constructivism has been also included as one of the possible ways in which social scientists interact and compete about (see Abbott's section referring to knowledge rediscovery). The description, questions and example of the first scenario have been amended to foster a sense of solidarity, mutual understanding and compassion to help inter-work.
Are there not pockets of ST practitioners that exhibit this competitiveness? Perhaps needs commenting on; particularly between hard and soft/critical systems thinkers (taking different epistemological stances on use of ‘systems’ idea – ontological vs nominal) but also, as you might be aware, amongst so called critical systems practitioners themselves (contingency approach of Jackson vs more universal - boundary critique - approach of Ulrich and Reynolds).
RESPONSE: Thank you for bringing this insight to light. The initial description of systems thinking has been enriched with tensions referring to paradigmatic and pluralism based discussions. The reference of Zhu (2011) has been added to refer to these tensions. The commonality of the transitions holism-pluralism from Reynolds and Holwell (2020) has been highlighted as a baseline from which the alternative pathway of thinking and action based on Abbott and Foucault's ideas is developed through the proposed scenarios.
So how might AST help in nurturing a better more ‘cooperative’ form of ‘correspondence’ between practitioner communities, other than rather flaky calling for more reflection, inclusion, and compassion? What is missing for me is some more sense of what is in the practice of AST that might help engagement with different perspectives in order to foster the transdisciplinarity needed as suggested by writers like Abbott. The example of mental health intervention is a good one. It is an example of problem structuring (not problem solving…) using SSM. So I think from the outset you need to highlight problem-structuring methods as distinct from problem-solving methods as being something of particular value in AST. But also there is a need for some more elaboration on why these methods are described as AST, and (albeit summarily) how they are administered, so as to make them, in your view, systems thinking methods. Again, for me, it’s the interplay between systematic and systemic that is core to contemporary tradition of STiP or AST (as distinct, for example, from traditions of systems science and complexity science, where the focus is on systematic analysis)
RESPONSE: Thank you for the suggestion. A more detailed description of each scenario together with how to use of systems approaches in each example has been provided
I think some more play on practitioner adaptability to different contexts and changing circumstances might be made, possibly along lines of bricolage. So a systems thinking practitioner can draw on many different traditions of practice in order to make relevant the interventions being constructed.
RESPONSE: The issue of adaptation has been highlighted in the amended positioning of systems thinking as a body of knowledge with tensions and opportunities.
“scientific articles published by prestigious journals (e.g. The Lancet® ) have quickly impacted government policies worldwide, only to be in some cases retracted and causing policy reversal.” …perhaps needs some reference for evidence.
RESPONSE: This sentence was rephrased and the use of digital technologies to support pandemic policies and responses has been added alongside it.
Section 4 (2nd scenario): I’m a little uneasy with notion that this ‘scenario’ is innovative and creative, suggesting by implication that scenario 1 is perhaps not (?). There is here though good emphasis on ethics (and I would suggest, perhaps more importantly, politics – with small ‘p’ – i.e. relations of power) counter-knowledge and possible need to preserve knowledges that are at risk of being further marginalised or even destroyed. Promoting practices like meditating, writing (journaling?), reflecting, and self-isolating, can appear to be a little fluffy without providing the traditions of (systems) practice on which they are, or can be, built. Here I’m thinking of Otto Sharmer’s work on presencing and Theory U, and traditions of ‘social learning’ (Chris Blackmore and Ray Ison). I wonder also if you might bring in the importance of marginalisation in AST as expressed through Churchman’s seminal (1979) publication The systems approach and its enemies’ (curiously not referenced), and of course the extension of this by Ulrich with CSH (see latest edition of this… in Ulrich, W. and Reynolds, M. (2020). Ch. 6. Critical Systems Heuristics: The Idea and Practice of Boundary Critique.)
RESPONSE: Thank you for this insight. Together with the reference of Reynolds and Holwell, a reference to Cordoba-Pachon (2006b) on the use of Foucault's ideas to reflect on power-ethics relations has been included and used throughout the presentation of both systems thinking as well as of the second scenario. The aforementioned practices of meditation and so on have been proposed and reflected upon in Cordoba-Pachon (2020). The focus on the individual regarding these practices has been maintained and signposted also in the second scenario example.
I like the idea of playful vigilance, but again wonder if this might also be applied to both scenarios?
RESPONSE: Thank you for this insight. Following it, the second scenario has been reworded as that of ethical vigilance and in relation to the world pandemic knowledge, measures and effects.
You may like to read Jon Harle blog posting cracks in the knowledge system: whose knowledge is valued in a pandemic and beyond? 28/08/2020. Which speaks to this paper rather nicely in my view.
RESPONSE: Thank you for this reference. It has been read and included in the proposed second scenario as it contributes to reinforce the importance of ethical vigilance in the light of power/knowledge relations about the world pandemic.
Reviewer 3 Report
This paper provided readers with a thorough and valuable analysis as well as insights into two different but complementary scenarios of applied systems and how they might evolve in the future of the new reality of the during and/or post pandemic world. It does raise a crucial issue of how to approach the pandemic altered societies and how to respond to new challenges brought to the surface by Covid-19 such as mental health, education, and others.
It is an invitation to further reflection and discussion among researchers, policy makers, managers and other stakeholders. The Author may want to expand his insights by providing a deeper understanding and detailed descriptions of the implementation processes needed to adopt the proposed scenarios in various societies.
Author Response
This paper provided readers with a thorough and valuable analysis as well as insights into two different but complementary scenarios of applied systems and how they might evolve in the future of the new reality of the during and/or post pandemic world. It does raise a crucial issue of how to approach the pandemic altered societies and how to respond to new challenges brought to the surface by Covid-19 such as mental health, education, and others.
It is an invitation to further reflection and discussion among researchers, policy makers, managers and other stakeholders. The Author may want to expand his insights by providing a deeper understanding and detailed descriptions of the implementation processes needed to adopt the proposed scenarios in various societies.
RESPONSE: THANK YOU FOR THESE VALUABLE INSIGHTS. AS MENTIONED IN THE RESPONSE TO OTHER REVIEWERS, THE PAPER HAS BEEN EXTENSIVELY REWRITTEN AND UPDATED TO GIVE FURTHER DETAIL ABOUT THE PROPOSED SCENARIOS, THEIR POSITIONING WITHIN EXISTING DEBATES IN SYSTEMS THINKING (THIS TERM WAS USED INSTEAD OF APPLIED SYSTEMS THINKING), THE QUESTIONS FORMULATED AND THE EXAMPLES. IN RELATION TO THE LATTER, MORE DETAIL WAS ADDED TO GUIDE SYSTEMS THINKERS WHO WOULD LIKE TO VENTURE IN PUTTING THE SCENARIOS IN PRACTICE AND BY USING SYSTEMS APPROACHES AND IDEAS. THE SCENARIOS NAMES WERE REWORDED TO ACCOUNT FOR THEIR COMPLEMENTARITY. AND THE CONCLUDING REMARKS WERE ALSO AMENDED TO INVITE SYSTEMS THINKERS TO ADOPT THEM TO HELP THEMSELVES AND OTHERS DEAL WITH (POST) PANDEMIC SITUATIONS.
Round 2
Reviewer 1 Report
The manuscript has been improved significantly. However, there are minor concerns that needs to be addressed by the authors.
- Line 295-296: what about causal loop diagram? This tool is an essential part of systems thinking to understand feedback loops and leverage points. How this tool can play a role in conveying the inter-work scenario?
- The last 2 paragraphs in Section 4.2 seems to be disconnected from the rest of the discussion. A question for the author: Why the discussion jumped from education to personal goal and activities? How this paragraph relates to ethical vigilance? Why the subject changes from 'he' to 'she' in this paragraph?
Author Response
- Line 295-296: what about causal loop diagram? This tool is an essential part of systems thinking to understand feedback loops and leverage points. How this tool can play a role in conveying the inter-work scenario?
RESPONSE: Thank you for this comment. The wording of the inter-work scenario was updated to reflect the possibility of using systems dynamics models and leverage points. The reference to Chapman (2004) was also added in this scenario to support this possibility.
- The last 2 paragraphs in Section 4.2 seems to be disconnected from the rest of the discussion. A question for the author: Why the discussion jumped from education to personal goal and activities? How this paragraph relates to ethical vigilance? Why the subject changes from 'he' to 'she' in this paragraph?
RESPONSE: Thank you for these comments. The paper was edited again in several sections and the paragraph in question was deleted. The second scenario content (features, implications, example) was amended to better link it with the first scenario. Also, the concluding remarks section was amended.